



# Establishment and preliminary application of forward modeling method for Doppler spectral density of ice particles

Han Ding[1], Liping Liu[1]

1 State Key Laboratory of Severe Weather, Chinese Academy of Meteorological Sciences, Beijing 100086, China

**Correspondence**: Liping Liu (liulp@cma.gov.cn)

**Abstract.** Owing to the various shapes of ice particles, the relationships between fall velocity, backscattering cross-section, mass, and particle size are complicated, which affects the application of cloud radar Doppler spectral density data to retrieve the microphysical properties of ice crystals. In this paper, under the assumption of six particle shape types, the relationships between particle mass, fall velocity, backscattering cross-section, and particle size were established based on existing research. Variations of Doppler spectral density with the same particle size distribution (PSD) of different ice particle types are discussed, and the radar-retrieved liquid and ice PSDs, water content, and mean volume-weighted particle diameter are compared with airborne *in situ* observations in Xingtai, Hebei Province, China, in 2018. The results showed the following: (1) for particles with the same equivalent diameter (De), the fall velocity of aggregates is the largest, followed by hexagonal columns, hexagonal plates, sector plates, and stellar crystals, with ice spheres falling two to three times faster than ice crystals with the same De. Hexagonal columns have the largest backscattering cross-section, followed by stellar crystals and sector plates, and the backscattering cross-sections of hexagonal plates and two kinds of aggregates are very close to those of ice spheres. (2) The width of the simulated radar Doppler spectral density generated by various ice crystal types with the same PSD is mainly affected by particle fall velocity and increased fall velocity rates with increased particle size, as do PSDs retrieved from the same Doppler spectral density data. (3) PSD comparisons showed that each ice crystal type retrieved from the cloud radar corresponded well to aircraft observations within a certain scale range when assuming that only a certain type of ice crystals existed in the cloud, which can fully prove the feasibility of retrieving ice PSDs from reflectivity spectral density.

## 1 Introduction

The importance of clouds in the Earth-atmosphere system is self-evident. Cloud microphysical



processes affect cloud distribution and lifetime, and ice clouds affect the Earth-atmosphere radiation balance by affecting long-wave and short-wave radiation transmission and change the atmosphere's thermodynamic structure Liou (1986). Further understanding of ice cloud microphysical properties has been clearly identified as essential to improving weather and climate change prediction (Delanoë et al.,

2005). The ice phase process is crucial to cloud and precipitation formation and development, and most precipitation in China is related to the ice phase process. Accurate particle size distribution (PSD) information is vital to precipitation and cloud radiation effect predictions in large-scale models. One of the most powerful tools for detecting non-precipitating and weakly precipitating clouds, the millimeter-wavelength cloud radar (MMCR), has short wavelength and high gain, which can effectively penetrate

cloud layers to continuously observe horizontal and vertical cloud structure changes under different dynamic conditions. This enables collection of more precise information on clouds.

  MMCR detects a Doppler spectrum that is a function of the backscattering cross-section and number of all particles in the radar detection volume with respect to their fall velocity. If the relationships between particle size and fall velocity and backscattering cross-section are known, Doppler spectral density data

can be used to obtain PSD characteristics of the cloud. However, radar-detected radial velocity includes particle fall velocity as well as vertical air motion, which is well known as one of the most difficult physical quantities to determine in meteorology. Identifying and eliminating air motion have presented difficulties in PSD retrieval using Doppler spectral density. If a small particle in a cloud (such as a liquid water droplet) is small enough, its fall speed relative to vertical air motion can be neglected, which means

that it can be used as a tracer to indicate clear-air motion (Kollias et al., 2001;Gossard, 1994). On this basis, it is possible to use Doppler spectral density data to retrieve cloud PSD.

  The relationships between fall velocity, backscattering cross-section, and particle size are easier to calculate for liquid particles because of their uniform shapes. Many studies have focused on raindrop size distribution retrieval using a Doppler spectrum. Liu et al. (2014) analyzed the accuracy of raindrop

size distributions retrieved from Doppler spectral density data observed by cloud radar. However, calculating fall velocity and backscattering cross-section for ice particles is complicated because of the complex shape of solid particles and fall velocity's sensitivity to changes in ambient temperature and humidity. This leads to many difficulties in interpreting and applying cloud radar data above the 0°C level. Yang et al. (2015) reviewed several classical computational approaches to light scattering





simulations of non-spherical ice crystals and discussed the strengths and weaknesses associated with

each approach. Liu (2008) built a database of microwave single-scattering properties for several non-

spherical particle shapes at frequencies from 15 to 340 GHz calculated using the discrete dipole

approximation method (DDA); however, he only calculated certain size particles rather than a

continuous relationship. In applied remote sensing, most previous studies have focused on the study of

PSD bulk characteristics and the relationship between PSD and microphysical parameters because of the

inability to obtain complete and specific PSD information, such as the relationship between radar

reflectivity $Z_e$, ice water content (IWC), and ice effective radius $r_e$, by means of remote sensing (Zhao

et al., 2016;Protat et al., 2007;Sassen et al., 2002;Mace et al., 2002). Zhong et al. (2012) indicated that

IWC retrieved using radar Doppler spectra is more reliable than IWC obtained using classic $Z_e$–IWC

relationships. So far, research on ice particle retrieval using MMCR Doppler spectral density in China

has not been found. Therefore, it is essential to establish the relationship between ice particle

microphysical parameters and Doppler spectral density data observed by radar and apply it to analyze

microphysical and dynamic properties.

    In this paper, we establish relationships between fall velocity, backscattering cross-section, and

particle size of six typical ice crystals based on a review of existing literature. Additionally, the

relationship between ice particle microphysical properties and radar-observed Doppler spectral density

is preliminarily explored. Data and methods are presented in the second and third sections of the paper.

Section 4 mainly analyzes relationships between particle microphysical parameters and particle size

based on calculation results, and Doppler spectra are simulated with the given PSD. Section 5 presents a

qualitative analysis of MMCR spectral density data and a brief comparison with aircraft observations.

Section 6 provides a discussion and conclusions.

## 2   Methods

### 2.1   Data

#### 2.1.1   Cloud Radar

From May 15 to 31, 2018, intensive cloud and precipitation observations were conducted at Huangsi

Meteorological Station in Hebei Province, China (114.21°E, 37.10°N). This experiment used a Ka-band

MMCR with a working frequency of 35 GHz and 8.6 mm wavelength. The MMCR operates in a

vertically pointing mode and has a vertical resolution of 30 m. A solid-state transmitter enables the radar



to make continuous observations. Four operational modes are applied to improve cloud and precipitation

radar detection capabilities: precipitation mode (M1), boundary mode (M2), middle level mode (M3), and cirrus mode (M4). Varying radar pulse widths and coherent and incoherent integration techniques were used to enable detection of low-level clouds and high-level weak clouds. During detection, the radar circulates through the four observation modes and converts reflected signals processed using fast Fourier transform into Doppler spectral data for storage. Table 1 shows the MMCR's major operational

parameters. Considering that particles above the zero layer have relatively small fall velocities and weak reflectivity, we use M3 mode data with high sensitivity and radial velocity resolution.

Table 1. Major parameters of the four operational modes.

| Item | Precipitation Mode (M1) | Boundary Mode (M2) | Middle Level Mode (M3) | Cirrus Mode (M4) |
|---|---|---|---|---|
| Pulse width | 0.2 μs | 2 μs | 8 μs | 20 μs |
| Pulse repetition frequency | 8000 Hz | 8000 Hz | 8000 Hz | 8000 Hz |
| Number of coherent integrations | 1 | 3 | 3 | 4 |
| Number of incoherent integrations | 4 | 4 | 4 | 4 |
| Number of fast Fourier transforms | 256 | 256 | 256 | 256 |
| Dwell time | 4 s | 4 s | 4 s | 4 s |
| Range sample volume spacing | 30 m | 30 m | 30 m | 30 m |
| Minimum range | 30 m | 300 m | 1200 m | 3000 m |
| Maximum range | 18 km | 18 km | 18 km | 18 km |
| Nyquist velocity | 17.13 m/s | 5.7 m/s | 5.7 m/s | 8.56 m/s |
| Velocity resolution | 0.134 m/s | 0.045 m/s | 0.045 m/s | 0.067 m/s |
| Minimum detective reflectivity at 5 km | −12.4 dBZ | −26.9 dBZ | −32.9 dBZ | −34.9 dBZ |

### 2.1.2 Aircraft instruments

From 5:17:42 to 5:49:35 UTC on May 21, 2018, stratiform precipitation occurred in the observation area. At the same time, an aircraft reached a maximum altitude of 4800 m and spiraled downward, with an approximately 18 km circling diameter, to 700 m around the observation site to observe cloud and precipitation vertical structures. The primary instruments in the aircraft included a modified cloud combination probe (CCP), a two-dimensional stereo probe (2D-S), a high-volume precipitation


spectrometer (HVPS), and an Aircraft Integrated Meteorological Measurement System (AIMMS-20),
which can provide meteorological data such as three-dimensional wind vectors, three-dimensional
aircraft position (i.e., latitude, longitude, and altitude), ambient temperature, and ambient relative
humidity. The CCP consists of a cloud droplet probe (CDP), a grayscale optical array imaging probe
(CIPgs), and a hotwire liquid water content sensor. Using a two-dimensional shadow cast technique, the
CIPgs detects cloud particles with diameters of 15–2000 μm. The 2D-S is an optical array imaging probe
that records projected areas of three-dimensional ice particles and PSDs from 10 to 1280 μm with
resolutions of 10, 20, 50, 100, and 200 μm. To mitigate the ice crystal shattering problem, we modified
probe tips and applied an arrival time algorithm to collected data to remove artifacts. HVPS data cover
150–47,075 μm with resolutions of 150 and 300 μm. Below this range, 2D-S data are used.

**2.2. Calculations of fall velocities, backscattering cross-sections, and ice particle mass**

When ignoring the effects of vertical air motion and turbulence, the relationship between the radar-
detected Doppler spectral density $S_Z(V_r)$ ( $mm^6 \cdot s \cdot m^{-4}$ ) and PSD $N(D_e)dD_e$ ( $m^{-3}$ ) can be
expressed as

$$S_Z(V_r) = CN\left(D_e(V_f)\right)\sigma\left(D_e(V_f)\right)\frac{\partial D_e(V_f)}{\partial V_f}, \tag{1}$$

where $C = 10^6\lambda^4|\varepsilon + 2|^2/(\pi^5|\varepsilon - 1|^2)$ is a constant related to the wavelength $\lambda$ and complex
permittivity $\varepsilon$ of precipitation particles and has a unit of $cm^4$. To compare and verify calculation
results, we equate ice crystal particles with different shapes to solid ice spheres of the same mass with an
equivalent diameter represented by $D_e$. $\sigma(D_e)(cm^2)$ is the particle's backscattering cross-section, $V_r$
(m/s) is the radar-detected radial velocity, and $V_f$ represents particle fall velocity (both $V_r$ and $V_f$
positive velocities are downward). Radar-observed radial velocity can only be regarded as particle
fall velocity in static air after air speed $V_a$ (positive speed is upward) is determined and eliminated
using the small particle tracing method (i.e., $V_f = V_r + V_a$). Particle size can then be calculated
based on the relationship between particle fall velocity and size. Thus, the relationship between
Doppler spectral density SZ and N(De) can be established based on the relationship between v-De
and σ-De. IWC (g·m⁻³) and mean volume-weighted diameter Dm (μm) can then be calculated after N(De)
is retrieved:

$$IWC = \frac{\pi\rho_{ice}}{6}\int N(D_e)\,D_e^3\,dD_e, \tag{2}$$

$$D_m = \frac{\int N(D_e)D_e^4\,dD_e}{\int N(D_e)\,D_e^3\,dD_e}, \tag{3}$$



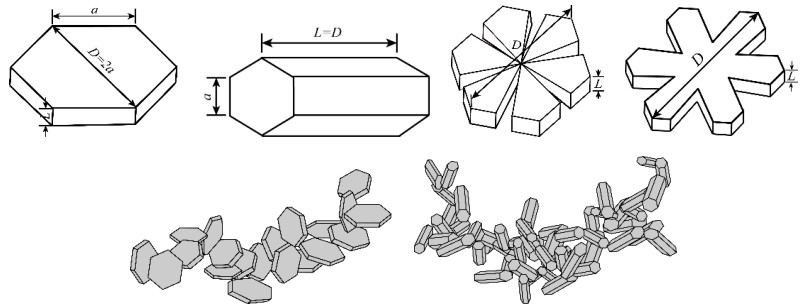


Figure 1. Ice particle geometries considered in this study.

From left to right: (top) hexagonal plate, hexagonal column, sector-like plates, stellar crystal with broad arms, (bottom) aggregate of plates, and aggregate of columns

Ice crystal fall velocity and scattering characteristics are known to be closely related to their shapes,

which are related to ambient temperature and supersaturation with respect to ice. MAGONO and Chung (1966) classified and named ice crystals found in nature as early as 1966. Over the past 70 years, researchers have developed many charts of ice crystal growth habits through laboratory research, in situ observation, or a combination of both. Most researchers agree that, when temperatures are greater than −18°C, the basic behavior of ice crystals goes from forming plates (0°C~−4°C), to columns (−4°C~−8°C),

to plates (−8°C~−22°C). Bailey and Hallett (2009) provided a detailed diagram of ice crystal growth habits by combining laboratory and field observations. They pointed out that ice crystal growth habits are dominated by various forms of polycrystals with two distinct habit regimes below −20°C: plate-like from −20°C to −40°C and columnar from −40°C to −70°C. On the basis of these results, this paper mainly discusses six typical ice particle types (four single crystal types and two aggregates). These types are

hexagonal plates (−8°C~−25°C, low ice supersaturation), hexagonal columns (below −40°C, ice supersaturation of 10%–25%), sector-like plates (−10°C~−25°C, high ice supersaturation), stellar crystals with broad arms (about −20°C, near water supersaturation), and plate and column aggregates. The six particle type shapes were thus chosen to represent complicated ice types (Fig. 1).

When calculating the backscattering cross-section of individual ice crystal particles, their shape

and fall attitude should be determined first. On the basis of the particle shape parameters given by Auer Jr and Veal (1970), the thickness of hexagonal and sector plates is $L = 2.02D^{0.449}$, that of stellar crystals is $L = 2.028D^{0.431}$, and the relationship between hexagonal column length L and half-width a is $a = 3.48L^{0.5}$ (Mitchell and Arontt, 1994). These measurements (a, L, and D) are all in μm. The





aspect ratio of the two kinds of aggregates is 0.6 (Hogan and Westbrook, 2014). Additionally, the

vertical orientation of all ice crystals is horizontal along the long axis when falling and horizontal

orientation is random.

### 2.2.1 Relationship between fall velocity and particle size

There are usually two ways to obtain the fall velocity $v_f$ of a single ice crystal particle: direct

laboratory measurements or field observations (mainly large-scale ice particles or snowflakes) and

calculation of the drag force based on aerodynamic principles. Fall velocity can then be calculated by

combining empirical relationships (such as mass and projected area-dimension power laws) and

measured or calculated results expressed in the form of an empirical power law (Matrosov and

Heymsfield, 2000). Obviously, results obtained through direct measurement are more accurate; however,

they are not applicable to particles of all scales, i.e., the application range is limited. Therefore, many

researchers have been working to determine a fall velocity equation based on particle size, mass, and

projected area (Mitchell and Heymsfield, 2005;Khvorostyanov and Curry, 2005, 2002;Mitchell, 1996).

Heymsfield and Westbrook (2010) evaluated the four most recent calculation methods of calculation at

that time and pointed out their shortcomings. They made a simple correction to the method proposed by

Mitchell (1996) to reduce sensitivity to area ratio and thus obtained a more accurate and simpler equation.

We used their proposed calculation method to calculate particle fall velocity. In general, when the force

of gravity on the particle is in balance with the drag force, $v_t$ can be calculated as follows:

(1) The modified best number can be calculated as $X^* = \frac{\rho_{air}}{\eta^2} \frac{8mg}{\pi A_r^{0.5}}$ when the values of $\eta$, $\rho_{air}$,

$m$, $A_r$, and D are given. Here, $\eta$ is the dynamic viscosity of air, $\rho_{air}$ is the density of air,

m is mass, and g is gravity. $A_r$ is defined as the particle's area ratio, i.e., the ratio of

projected area (A) to the particle's circumscribed circle area. The m- and A-D relations are

obtained using the formula compiled by Mitchell (1996), and the coefficients used are given in

Table 2. Here D is the particle's maximum diameter (the diameter of the circumscribed circle

of the particle).

(2) Then, the Reynolds number is estimated as $R_e = \frac{\delta_0^2}{4}\left[\left(1 + \frac{4\sqrt{X^*}}{\delta_0^2\sqrt{C_0}}\right)^{1/2} - 1\right]^2$ using $C_0 = 0.35$

and $\delta_0 = 8.0$.

(3) Finally, particle fall velocity is directly computed as $v_f = \frac{\eta R_e}{D\rho_{air}}$.



Table 2. Coefficients of mass- and area-dimensional power laws for six ice particle types
($m = aD^b$, $A = \alpha D^\beta$, units are cgs)

| Particle type | Mass | | Area | |
| --- | --- | --- | --- | --- |
| | $a$ | $b$ | $\alpha$ | $\beta$ |
| *Hexagonal plates* | | | | |
| *100 μm ⩽ D ⩽ 3000 μm* | 0.00739 | 2.45 | 0.65 | 2.0 |
| *Hexagonal columns* | | | | |
| *30 μm < D ⩽ 100 μm* | 0.1677 | 2.91 | 0.684 | 2.0 |
| *100 μm < D ⩽ 300 μm* | 0.00166 | 1.91 | 0.0696 | 1.50 |
| *D > 300 μm* | 0.000907 | 1.74 | 0.0512 | 1.414 |
| *Crystals with sector-like branches* | | | | |
| *10 μm < D ⩽ 40 μm* | 0.00614 | 2.42 | 0.24 | 1.85 |
| *40 μm < D ⩽ 2000 μm* | 0.00142 | 2.02 | 0.55 | 1.97 |
| *Stellar crystals with broad arms* | | | | |
| *10 μm ⩽ D ⩽ 90 μm* | | | | |
| *90 μm < D ⩽ 1500 μm* | 0.00583 | 2.42 | 0.24 | 1.85 |
| | 0.00027 | 1.67 | 0.11 | 1.63 |
| *Aggregates of plates* | | | | |
| *600 μm ⩽ D ⩽ 4100 μm* | 0.0033 | 2.2 | 0.2285 | 1.88 |
| *Aggregates of columns* | | | | |
| *800 μm ⩽ D ⩽ 4500 μm* | 0.0028 | 2.1 | 0.2285 | 1.88 |

**2.2.2 Relationship between backscattering cross-section and particle size**

After establishing the relationship between particle fall velocity and particle size, it is necessary to
know the relationship between backscattering cross-section and size to determine the PSD using Doppler
spectral density. Because the ice crystal shapes are complex, it is essential to establish a scattering model
using a simple theory, a simple calculation, and reliable results to obtain the backscattering characteristics
of non-spherical ice crystals. One approach is to simplify particle shapes by treating them as spheres or
ellipsoids and then apply a suitable scattering theory. For example, spheres can be calculated using the
Mie scattering theory (Mie, 1908), and ellipsoids can be calculated using the T-matrix method. Although
the T-matrix method can effectively calculate the scattering characteristics of spherical particles (Macke
et al., 1996), it can only be applied to a limited particle size range, beyond which numerical stability
problems are likely to occur during calculation. This makes processing complex when calculating





particles of different shapes, so it is difficult to apply to the scattering calculation of particles of complex shapes. Another method is to use a simplified scattering theory, such as the Rayleigh–Gans approximation theory (RGA), rather than simplified particle shapes (Craig et al., 1983). RGA ignores internal higher-order interactions of electromagnetic waves, greatly simplifying mathematical problems.

The RGA method is adopted in this paper. In RGA theory, the backscattering cross-section of particles with arbitrary shapes to plane waves in the s direction can be written as

$$\sigma_b = \frac{9k^4|K|^2}{4\pi}\left|\int_{-D/2}^{D/2} A(s)\exp(i2ks)\,ds\right|^2, \tag{4}$$

where k is the wave number, $K = (\varepsilon - 1)/(\varepsilon + 2)$, $\varepsilon$ is the complex permittivity of ice, and A(s) is the area of ice crystals intersected by the plane. Hogan and Westbrook (2014) made some

improvements to the RGA method in 2014, creating a self-similar Rayleigh–Gans approximation (SSRGA) that can be used to calculate the backscattering cross-section of aggregates. Although aggregate structures are complex and difficult to predict, Hogan et al. found that the aggregation process of crystals is self-similar and can be described by a power law. Thus, they proposed an equation to calculate the backscattering cross-section of particles in the centimeter and millimeter

bands:

$$\sigma_b = \frac{9\pi k^4|K|^2 V^2}{16}\left\{\cos^2\left[\left(1+\frac{\kappa}{3}\right)\left(\frac{1}{2x+\pi}-\frac{1}{2x-\pi}\right)-\kappa\left(\frac{1}{2x+3\pi}-\frac{1}{2x-3\pi}\right)\right]^2\right.$$

$$\left.+\beta\sum_{j=1}^{n}(2j)^{-\gamma}\sin^2(x)\left[\frac{1}{(2x+2\pi j)^2}-\frac{1}{(2x-2\pi j)^2}\right]\right\}, \tag{5}$$

Here, x = kD, $\kappa$ is the kurtosis parameter, and $\beta$ is the prefactor of the power law. Hogan and Westbrook (2014) also point out that, when ice crystals collide into aggregates, the overall shape of

the aggregates affects scattering properties rather than the shape and size of individual crystals making up the aggregates. A(s) of six types of ice crystals are calculated using the parameters given in Table 2, the backscattering cross-sections of four kinds of single ice crystals are calculated using Eq. (4), and the backscattering cross-sections of two kinds of aggregates are calculated using Eq. (5). The kurtosis and power-law prefactor parameters can be found in Hogan and Westbrook (2014).

**2.3 Analysis and simulation**

**2.3.1 Fall velocities and backscattering cross-sections of different ice particle types of ice particles**





To establish the relationship between Doppler spectral density data and PSD, we calculated the masses, fall velocities, and backscattering cross-sections of ice crystals with different sizes based on the methods mentioned in Section 2.2. Figure 2a shows the relationship between mass and maximum

dimension of six ice particle types within physical size limits. Figure 2b shows the corresponding relationship between maximum diameter and ice sphere equivalent diameter of different particles. These figures show that, when maximum diameter is same, hexagonal plates have the largest mass, followed by the two aggregate types and sector plates, with the stellar plate crystals having the smallest mass. The masses of the two aggregate types are almost equal when D is less than 2500 μm, and the masses of the

sector plates are similar to those of hexagonal columns when D is less than 2000 μm.

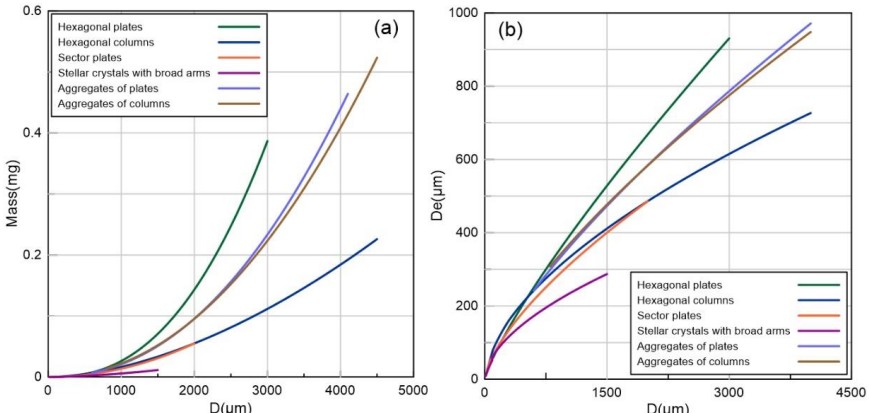

Figure 2. The relationship between mass and maximum diameter of ice crystals (a) and the corresponding relationship between maximum diameter and equivalent ice sphere diameter (b)

Because the air viscosity coefficient mainly depends on air density, air density changes will affect

particle fall velocity. Thus, even if particles are of the same size, fall velocity will change with height. We take the fall velocity of particles at a height of 4.5 km under a standard atmosphere as an example for analysis. Figure 3a shows the $v_f$–De relationship of ice crystals and an ice sphere. An ice sphere falls approximately 2–3 times faster than ice particles with the same De. Among the six ice crystal types, the two aggregates have the greatest fall velocity, followed by hexagonal columns, hexagonal plates, sector

plates, and stellar plates, which have the smallest fall velocity. Because of size limitations, sector-like plate and stellar plate crystals have relatively small maximum fall velocities, which slowly increase with particle size. By contrast, the fall velocity of hexagonal columns increases fastest with increasing size.



At the same time, if we suppose that all ice particles in a cloud are spherical, ice particle sizes

corresponding to the same fall velocity will be much smaller than actual ice particles of different shapes.

This will impose serious errors on the following calculation, causing it to deviate from the real situation

when retrieving PSD spectra using radar data.

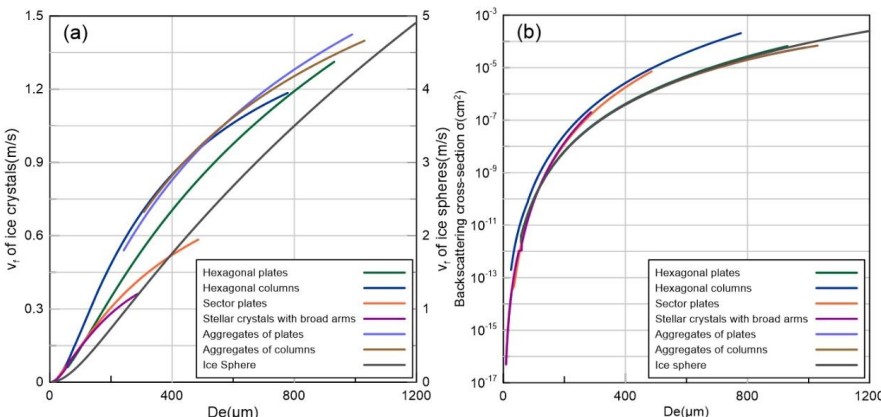

Figure 3. Relationships between the fall velocities of different types of ice crystals and an ice sphere (a)

and the backscattering cross-section of ice crystals and an ice sphere (b).

Fall velocity of particles at a height of 4.5 km under standard atmospheric conditions.

The backscattering cross-sections of single ice particles and aggregates were calculated using the

method mentioned above, and the results are shown in Fig. 3b. When ice crystals are small, the

differences in backscattering cross-sections of different ice crystal types are also small. For the same De

values, hexagonal columns have the largest backscattering cross-section, followed by stellar plates, sector

plates, hexagonal plates, both aggregate types, and ice spheres, which have the smallest backscattering

cross-section. However, the backscattering cross-sections are all almost equal. Additionally, we found

that backscattering cross-section size has little relation with shapes of particles with the same volume.

Assuming that two crystals of the same mass have the same density (i.e., they have the same volume)

and the projected area is large or small (i.e., thickness is thick or thin), the backscattering cross-section

difference between them is very small and can thus be neglected. In other words, for ice crystals of the

same mass, backscattering cross-section size depends only on the integral of the projected area in the

electromagnetic wave propagation direction. Therefore, it is crucial to choose shape parameters for ice

particle types, which will significantly affect the calculation results. However, ice crystal shapes are very



complex; thus, it is obvious that we could not cover all of them in the calculation. Therefore, we can only

select typical scale parameters to obtain a more representative relationship between backscattering cross-

section and particle size.

### 2.3.2 Doppler spectral density and PSD retrieval simulations

From the above results, if PSD is given, the Doppler spectrum can be calculated based on Eq. (1).

Many studies have proposed various functions to describe the PSD of ice crystals, such as the negative

power function (Heymsfield and Platt, 1984), the power function (Platt, 1997), and the Gamma function

(Ryan, 2000). Gunn and Marshall (1958) used the exponential function to describe ice crystal PSD, which

has been widely used since then:

$$N(D_e) = N_0 \exp(-\lambda D_e), \tag{6}$$

$$Z_e = \int_0^v S_z(v_f)dv, \tag{7}$$

Here, $N(D_e)$ ($m^{-3}$ $mm^{-1}$) is the number of particles per unit volume per unit, $N_0$ ($m^{-3}$ $mm^{-1}$) is the

intercept parameter, and $\lambda$ ($mm^{-1}$) is the shape parameter. Wang et al. (2015) gave statistical results

of ice crystal PSD in East Asia, with average $N_0$ and $\lambda$ values of 2035.1 $m^{-3}$ $mm^{-1}$ and 1.23 $mm^{-1}$,

respectively. For single ice crystals, we assume a De smaller than 287 μm (this scale was chosen

because of physical size limitations of stellar crystals), and the average PSD was used to calculate the

Doppler density produced by four kinds of single ice crystals (based on Eq. (1)). For the two aggregate

types, we calculated the Doppler spectral density with De values of 307–989 μm. The PSDs used for

simulation are shown in Fig. 4a. By using Eq. (7), the equivalent reflectivity values generated by the six

crystal types are −9.76, −1.17, −4.08, −0.51, 25.65, and 25.92 dBZ (hexagonal plates, hexagonal columns,

sector-like plates, stellar crystals, aggregates of plates, aggregates of columns). It is important to note

that the ice crystals and aggregates have different sizes. The different shapes of the four ice crystals

introduce a reflectivity bias of 9.25 dBZ. However, the two kinds of aggregates have similar reflectivity

values.



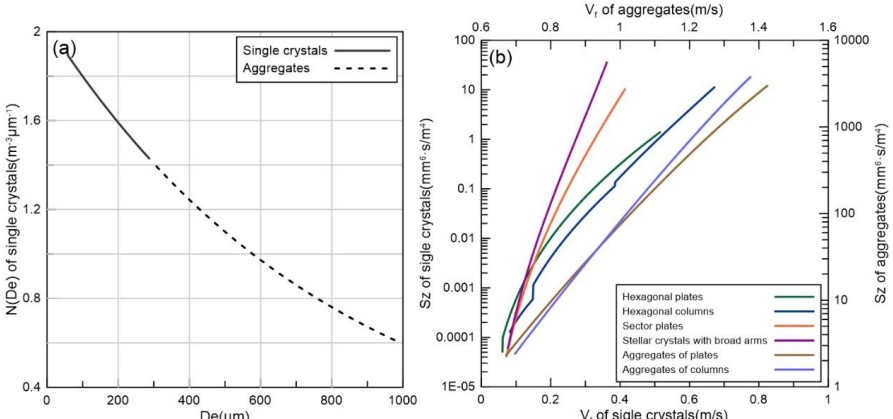

Figure 4. Exponential distribution used in this paper (a) and the Doppler spectral density calculated

from the exponential distribution (b)

First, single ice crystals are compared. As shown in Fig. 4b, for the same PSD, Doppler spectral density width is mainly inversely proportional to fall velocity and fall velocity growth rate. The faster the ice crystal fall velocity increase with particle size, the wider the generated Doppler spectrum and vice versa. Figure 3a shows that, for the same De, the fall velocity increases fastest for hexagonal columns, followed by hexagonal plates, sector plates, and stellar plates, which have the slowest increase. Thus, hexagonal columns have the widest Doppler spectrum, followed by hexagonal plates, sector plates, and stellar plates, which have the narrowest Doppler spectrum. The Doppler spectrum depends on particle backscattering cross-section, as shown in Eq. (1), and SZ is jointly determined by $\sigma$ and $\partial D / \partial V_f$, which is proportional to the backscattering cross-section and inversely proportional to the rate of velocity change with particle size. Stellar plates have the largest spectral reflectivity among the single ice crystal types, followed by hexagonal columns, sector plates, and hexagonal plates, which have the smallest spectral reflectivity. The microphysical relationship that we have established combined with the fact that stellar crystals have very small $\partial D / \partial V_f$ values means that, although stellar crystals do not have the largest backscattering cross-section, they do have a large Doppler spectral density. Additionally, even if the fall velocity is small and the Doppler spectrum is narrow, it has the biggest Ze of single ice crystals. Hexagonal plates have the lowest reflectivity spectral density because of their small backscattering cross-section and large $\partial D / \partial V_f$ value. The Doppler spectrum width and intensity of the two aggregates are similar and much larger than those of single ice crystals. Aggregates of columns have a slightly higher Doppler spectral density than aggregates of plates.



Further, it is easy to find that the relationship of SZ between ice crystal types and that the relationship of Ze is completely consistent, which means that, although the same PSD has a different Doppler spectrum width because of the different fall velocities of different crystal types, the magnitude of Ze mainly depends on Doppler spectrum intensity magnitude.

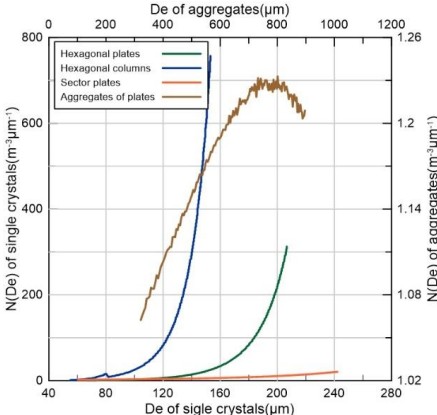

Figure 5. Retrieved PSDs for four ice particle types with the same SZ

The Doppler spectra of stellar plates and aggregates of columns obtained by the simulation were used to derive the PSD of other single ice crystal types and aggregates of plates, respectively, to analyze the effects of ice type variations on retrieved microphysical parameters. Note that single ice crystals and aggregates have different coordinate axes to show the inversion results in Fig. 5. When De > 80 μm, the

retrieved PSDs for hexagonal plates, hexagonal columns, and sector plates are obviously larger than the original one. The differences in retrieved PSDs between sector plates and other single ice crystal types are mainly due to the differences in fall velocity. Compared to other single ice crystal types, sector plates have the largest De at the same fall velocity, yielding a larger backscattering cross-section, which, when coupled with a slower increase in velocity, results in a low concentration. The hexagonal plates and

hexagonal columns have smaller De values (and smaller backscattering cross-sections) at the same fall velocity as well as a faster increase in velocity, which leads to a higher concentration. The width and value of the retrieved PSD of aggregates of plates are both similar to those of aggregates of columns because they have similar fall velocities and backscattering cross-sections.

### 3 Results and discussion

#### 3.1 Retrieved results from MMCR

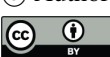



Reflectivity and Doppler spectral density data from M3 mode, with higher speed resolution and sensitivity, were used to retrieve PSD and IWC. The high-velocity resolution means that a fine PSD can be produced. On the basis of the relationships that we have established, more sophisticated Doppler spectra will yield more accurate PSDs. Figure 6a shows raw Doppler spectral density data of a beam

obtained at 5:38 (UTC) by MMCR. The SZ with negative velocity is due to velocity aliasing, and the side lobe echo is highlighted with a red circle. Algorithms for dealiasing singly wrapped aliased Doppler spectral density and for detecting and removing artifacts produced by pulse compression were used on the raw SZ (Liu and Zheng, 2019). Reflectivity and retrieved air speed profiles based on "small-particle tracers" are shown in Fig. 6b. On the basis of the Ze profile and Doppler spectrum, we inferred that the

melting layer is at approximately 4.15–4.5 km. Additionally, the Doppler spectrum width suddenly narrows at 4.5 km, indicating that echoes above this height are mainly generated by ice particles. Below the bottom of the melting layer (4.15 km), we inversed the Doppler spectrum using the v–D relation of raindrops given by Gossard (1994) and the σ–D relation calculated using the extended boundary condition method (Barber and Yeh, 1975). Above the top of the melting layer (4.5 km), we assumed only

one ice crystal type in the cloud and used the microphysical relations established in Section 3 to derive the PSD. In addition, the size of all ice particle types was controlled within the limit of particle physical scale.

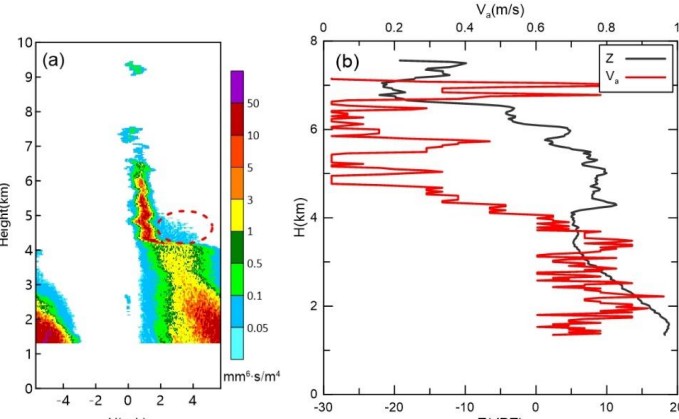

Figure 6. Doppler spectrum obtained by M3 at 5:38 (UTC) (a) and reflectivity and vertical air speed (b)

Generally, the change trends of PSD widths of the six particle types are similar to those of the Doppler spectral width with height. However, the concentrations of different types of particles retrieved





from the same height are different. For Doppler spectral density data, the corresponding velocity represents particle size; thus, the droplet spectrum width is mainly determined by particle fall velocity. In other words, the smaller the particle size corresponding to the same velocity, the more left broadened

the derived PSD spectral width. Similarly, the slower the fall velocity increases with particle size, the larger the particle size corresponding to the large fall velocity and the more right broadened the derived PSD spectral width. By comparing the PSD results of the six particle types, we can see that the PSD spectra are incomplete because of the limitation of particle physical scale (only a part of them exist). The hexagonal plates and two kinds of aggregates are mainly affected by the lower limit of the physical scale,

and their minimum discernible $De$ values are 70, 240, and 300 μm, respectively. Additionally, the hexagonal columns, sector plates, and stellar plates are mainly affected by the upper limit of physical scale, with maximum discernible $De$ values of 800, 500, and 300 μm, respectively. There must be no aggregates of columns in the inversion results when $De$ is less than 300 μm, and aggregates must be present when $De$ is greater than 930 μm. It is important to note that the physical scale constraint does not

allow all Doppler spectra density data to be involved in PSD retrieval when only one ice particle type is assumed to be present in the cloud. That is also the reason why the $Ze$ values calculated using the PSD retrieved for different ice crystals are slightly less than the reflectivity values shown in Fig. 6b.

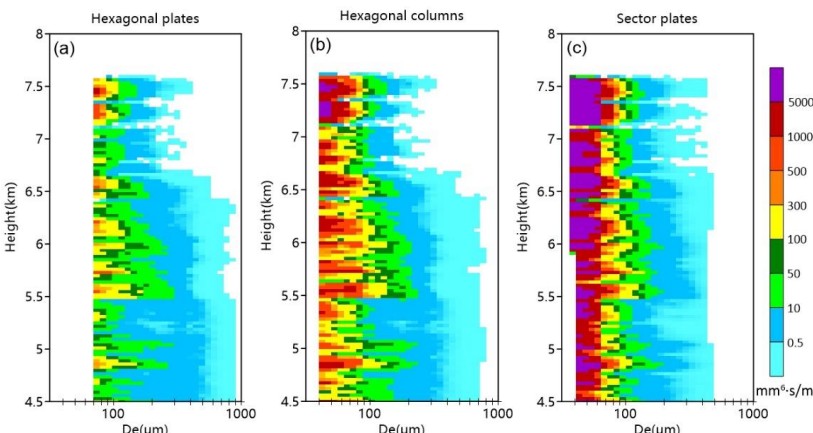





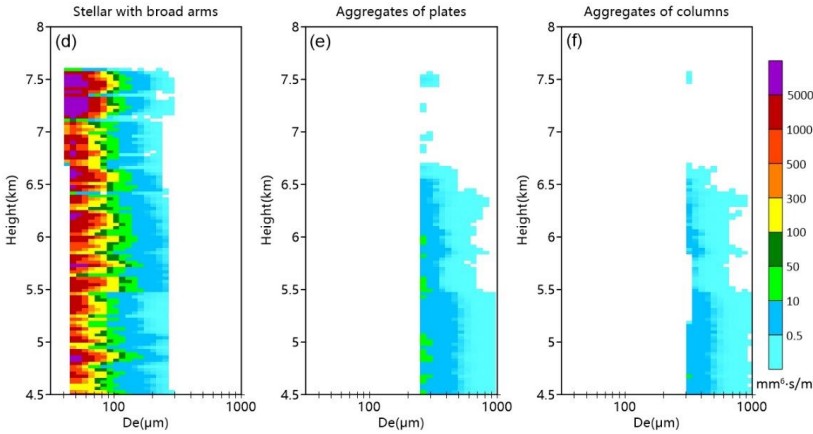

Figure 7. Retrieved PSDs at different heights for six ice crystal types

Moreover, the concentration of small particles is largest at each height, and particle concentration decreases with increasing particle size. The conclusion of the simulation demonstrated that particle number is inversely proportional to the particle's $\sigma$ and directly proportional to $\partial D / \partial V_f$. Sector plates have the highest concentration when De is less than 100 μm and have a smaller De than stellar crystals for the same falling speed; thus, the smaller backscattering cross-section coupled with a larger $\partial D / \partial V_f$ results in a high concentration. Although the two kinds of aggregates only have the PSD of large-size particles, the aggregates of plates have a slightly higher concentration than that of column aggregations. Because the two have almost identical backscattering cross-sections, the difference is mainly caused by the fall velocity. Considering the existence of echo generated by particles with small speed, there must be single crystals within the radar detection volume.

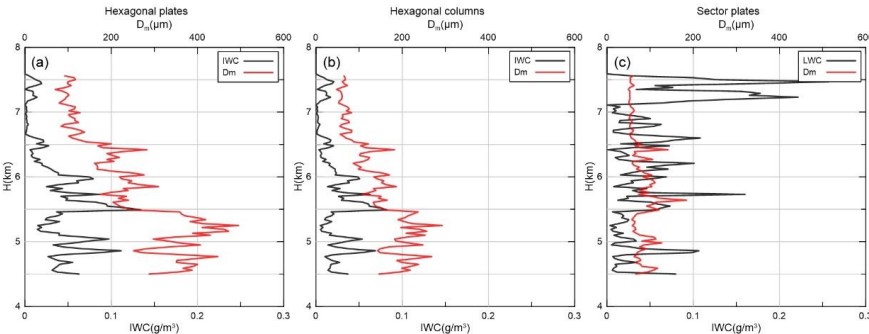





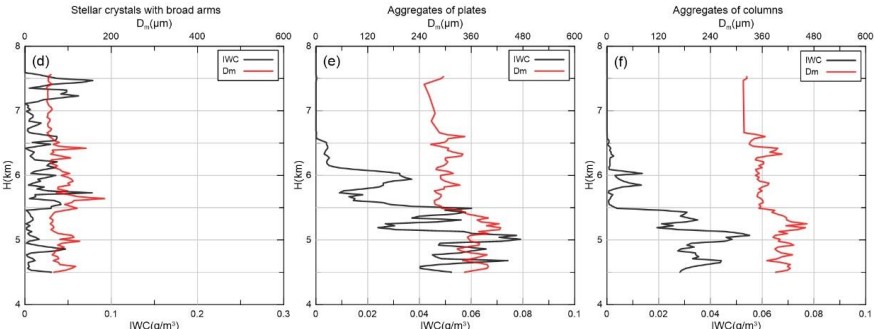

Figure 8. Mean volume-weighted diameter and IWC of different ice crystals retrieved from radar data

Figure 8 shows retrieved IWC and Dm profiles for the six ice particle types. IWC values derived

from hexagonal plates and hexagonal columns show a consistent trend with height, with relatively

uniform changes below 6 km and gradual decreases with increasing height above 6 km. Because of

physical size limitations, hexagonal plates have a narrower PSD spectrum than hexagonal columns.

However, the small particle concentration for hexagonal plates is smaller than that of hexagonal columns,

and the concentration of large particles is larger; thus, hexagonal plates have an IWC of about 0–0.12

g/m³, whereas the IWC of hexagonal columns is approximately 0–0.08 g/m³. Additionally, sector plates

and stellar plates have relatively consistent IWC change trends, with overall changes relatively uniform

below 7 km, and IWC increasing sharply above 7 km. Figure 7 shows that sector plates have wider PSD

spectra and a larger concentration; thus, their IWC is larger, ranging 0–0.1 g/m³ below 7 km. Stellar

crystals have an IWC of approximately 0–0.04 g/m³, which is roughly half that of the sector plates. The

IWC variation trends of the aggregates are also very similar, although they only have the PSD spectra of

large particles as the large particle concentration is slightly larger than that of other ice crystals, especially

for aggregates of plates. The aggregates of columns and plates have IWC values that are not particularly

small, at 0–0.08 and 0–0.06 g/m³, respectively. Additionally, the Dm profiles of different ice crystals are

similar to the PSDs shown in Fig. 7. The more the PSD spectrum widens to the right, the bigger the Dm

will be, and the narrower the PSD spectrum becomes to the left, the smaller the Dm will be. Hexagonal

plates have Dm values about 1.5 times larger than those of hexagonal columns, mainly because of the

lack of small particles in the hexagonal plates' PSD. The Dm values of the aggregates are similar, and

the Dm values of sector plates and stellar plates are almost identical. Additionally, the change of all Dm

values with height is similar to changes in IWC.



**3.2   Comparison with aircraft detection**

Because the aircraft's maximum flight altitude is about 4900 m, to compare the inversion results of MMCR with the PSD observed by the aircraft, the particle concentration above 4.2 km observed by the 2D-S and CIP probes was first averaged and then compared with the PSD retrieved from MMCR data at 4.5 km. The results are shown in Fig. 9 (note that both the aircraft and radar results are denoted by the

maximum particle size because of the way the probe takes measurements). The 2D-S and CIP probes have consistent measurements. The particle concentration decreases with increased particle size, and the concentration of 400–1200 μm particles detected by two probes was almost equal. Compared with the radar inversion results, the trend in concentration variation with particle size was roughly the same; however, the radar inversion concentration decreases more quickly with increased particle size and was

much larger than that observed by aircraft for small particles. Focusing on the concentration of particles smaller than 2000 μm, the size ranges of different ice crystal types with almost identical concentrations as observed by aircraft were as follows: hexagonal plates (800–1400 μm), hexagonal columns (600–1000 μm), sector plates (400–600 μm), stellar plates (400–600 μm), aggregates of columns (1200–1800 μm), and aggregates of plates (1200–1800 μm). In fact, because of the aircraft probe's small sampling volume

and the large range bin observed by radar, inconsistencies in particle concentration are inevitable. Additionally, precipitation occurred during the joint observation of the aircraft and radar; thus, cloud microphysical processes changed rapidly over time, and the aircraft data's time (altitude) average also led to some concentration differences between observation and reality.

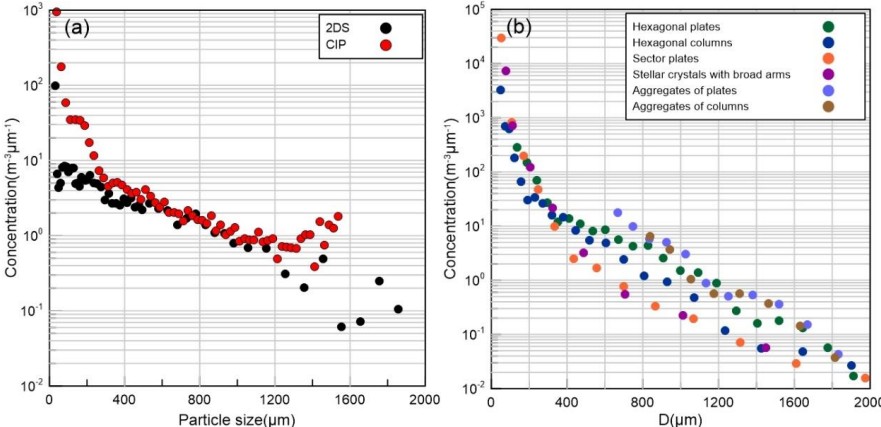

Figure 9. Particle size distribution observed by 2D-S and CIP probes (a) and retrieved by MMCR at 4.5 km (b)



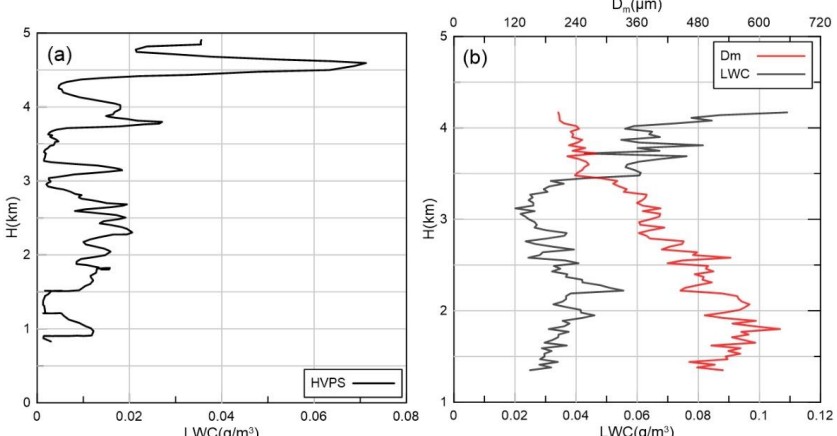

Figure 10. Liquid water content (LWC) observed by HVPS (a) and calculated from PSD obtained by

MMCR (b)

Using the same retrieval method, the liquid hydrometeor PSD was retrieved from Doppler spectra density data below 4.2 km based on microphysical relationships of liquid droplets. Then, liquid water content (LWC) and Dm were calculated using Eqs. (2) and (3). Figure 10a shows the LWC detected by the aircraft's HVPS probe, and the LWC and Dm values in Fig. 10b were retrieved from MMCR. The trend of LWC with height of the two is basically the same, with aircraft-observed LWC within 0–0.02

g/m$^3$ and the radar-retrieved results within 0.02–0.05 g/m$^3$ below 3 km. Above 3 km, both aircraft-obtained and radar-retrieved LWC values increase with height; however, radar-based LWC results are still larger than those of the aircraft overall. It is worth noting that the HVPS has a detection range for particles larger than 75 µm and has a 150-µm resolution, making it inevitable that some small particles were missed, resulting in the underestimation of LWC. Additionally, Dm increases with height below 2

km and decreases with height above 2 km, with the peak Dm value appearing at 1.7 km. Figure 7a also shows that the most severe aliasing occurs at 1.5 km.





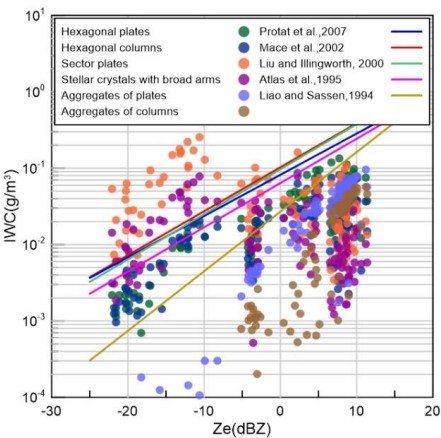

Figure 11. Relationship between Ze and ice water content (IWC)

Many studies have given coefficients of the statistical relation between Ze and IWC. We chose

coefficients from several studies to compare against our MMCR-derived IWC results. The relationships

between Ze (mm$^6$ m$^{-3}$) and IWC (g/m$^3$) of ice clouds used in this paper are listed in detail in Table 3.

Reflectivity above 4.5 km was used to calculate IWC based on five Ze-IWC values. As shown in Fig. 11,

hexagonal plates and hexagonal columns have IWC values very close to the results from Atlas et al.

(1995) and slightly less than the results from Protat et al. (2007), Mace et al. (2002), and Liu and

Illingworth (2000). Lower Ze values led to the underestimation of IWC for the two kinds of aggregates,

and degree underestimation decreased with increasing Ze. The IWC retrieved from sector plates and

stellar plates when Ze values were less than −5 dBZ was larger than that calculated from other studies

and was less when Ze values were greater than −5 dBZ. Almost all of the ice crystal types have IWC

values less than those from other studies when the reflectivity was stronger than 0 dBZ, and this

underestimation tends to become more serious when reflectivity was greater than 10 dBZ.

Table 3. Statistical empirical parameters for ice clouds using Ka-band MMCR observations

| Reference | Equation | Cloud type |
|---|---|---|
| Liao and Sassen (1994) | IWC $= 0.027Z_e^{0.78}$ | Ice cloud |
| Atlas et al. (1995) | IWC $= 0.064Z_e^{0.58}$ | Ice cloud |
| Liu and Illingworth (2000) | IWC $= 0.097Z_e^{0.59}$ | Cirrus cloud |
| Mace et al. (2002) | IWC $= 0.103Z_e^{0.576}$ | Cirrus cloud |
| Protat et al. (2007) | IWC $= 0.082Z_e^{0.54}$ | Mid- and high-latitude ice cloud |

## 4   Conclusions


To use Ka-band cloud radar Doppler spectral density data to quantitatively analyze the microphysical

and dynamic structure properties of cloud ice processes, this paper established the relationship between

fall velocity, backscattering cross-section, and particle size of six typical ice crystal types, analyzed the

microphysical properties of various particles and their influence on cloud radar reflectivity and Doppler

spectral density, and applied the established relationship to the retrieval of ice microphysical properties

using Doppler spectral density data on the east side of Taihang mountain in Hebei Province, China. We

then obtained PSDs at different heights and compared them with the results from aircraft observation.

The main conclusions of this paper are as follows:

(1)    For particles with the same equivalent diameter, the fall velocity of aggregates is the largest,

followed by hexagonal columns, hexagonal plates, sector plates, and stellar crystals. Hexagonal columns

have the largest backscattering cross-section, followed by stellar crystals and sector plates, and the

backscattering cross-sections of hexagonal plates and the two aggregate types are very close to those of

ice spheres. However, ice spheres fall two to three times faster than ice crystals with the same De, which

means that considering different ice crystal types as ice spheres while retrieving the PSD using Doppler

spectral data will lead to different degrees of particle scale underestimation.

(2)    The spectral width of the radar Doppler spectrum generated by the same PSD is mainly

affected by particle fall velocity, with an increasing rate of fall velocity with increased particle size. The

faster fall velocity increases with particle scale, the narrower the Doppler spectrum is. And the reflectivity

intensity is determined by particle backscattering cross-section and the rate of velocity change with scale,

which is inversely proportional to the particle backscattering cross-section value and directly

proportional to the rate of fall velocity increase. Additionally, differences between different PSDs

retrieved from the same Doppler spectral density data are mainly caused by the fall velocity.

(3)    We assume that only a certain type of ice crystal existed in the cloud, and the PSD comparison

showed that the radar inversions of each ice crystal type corresponded well to aircraft observations within

a certain scale range, indicating a great possibility of the existence of this type of particle within that

range and fully verifying the feasibility and reliability of ice PSD retrieval from Doppler spectra. The

LWC variation trend with height between radar inversion and aircraft observation was basically the same,

whereas the aircraft-measured value was slightly smaller than that of radar inversion. Additionally,

precipitation occurred during aircraft observation, leading to rapid changes in cloud microphysical



structure over time, and height average was carried out to facilitate the comparison with aircraft observations, making the situation different from reality, which made comparison with radar results more

difficult.

The above work is a preliminary attempt to establish a forward modeling method for Doppler spectral density data of solid precipitation particles. In the future, more microphysical parameters of precipitation particles can be used to establish a more complete relationship and aid in the interpretation and analysis of radar Doppler spectral density data. In addition, this paper uses one set of shape

parameters for the calculation of each ice crystal type; however, there are many other ice crystal proportions in nature. At the same time, when retrieving the PSD using Doppler spectral density data, we assumed that only one-type particles exist in the cloud, which is obviously inconsistent with reality as the actual type and distribution of ice particles in a cloud are complex. Assuming that there is only one particle type in a cloud is not consistent with the actual situation. The actual distribution of ice crystal

particle types is very complex, and subsequent simulation and inversion can be carried out with a mix of all kinds of ice crystals in proportions based on observations, bringing it closer to the real environment. In short, with such a set of microphysical parameters and size relations of various ice crystal particles, more Doppler spectral density data can be analyzed to statistically study the solid microphysical properties of different areas (such as ice clouds on the Tibetan Plateau).


*Author contribution*. Conceptualization and methodology, L.L.; Formal analysis and software, D.H.

*Acknowledgements*. This research was funded by the National Natural Science Foundation of China (91837310 and 41875036).

*Competing interests*. The authors declare that they have no conflicts of interest.

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
