# Peer review of "Establishment and preliminary application of forward modeling method for Doppler spectral density of ice particles"

_Atmospheric Measurement Techniques, 2019_

## Referee Comment (RC1) · Anonymous Referee #1 · 26 Sep 2019

The authors present a forward modelling approach for six different ice particle types, with a particular focus on the Doppler radar spectral density. Starting from six different, literature-based particle descriptions of mass-, area- and velocity-size relationships and by using self-similar Rayleigh-Gans approximation to determine the backscattering cross section of the particles they build a forward model for each particle size and type. This is initially used in combination with an assumed particle size distribution (inverse exponential) to evaluate the relative differences in the forward-modelled Doppler radar spectral density for the different particle types. Later this method is inverted using radar observations to estimate the particle size distribution in clouds observed over China, which are additionally evaluated against aircraft-based measurements of the

particle size distribution. The retrieved particle size distributions (one for each particle type) show an uncertainty of around one order of magnitude, but are consistent with the aircraft-measured particle size distribution.

In general I find both the paper and the results presented interesting. However, in several parts I found the description of the method either confusing or incomplete. The figures are generally well chosen and clear, although some can be improved (see below). The method applied seems reasonable (although without further clarification of exactly what was done, I cannot be sure) and the agreement of the retrieved size distribution with aircraft observations is encouraging. The authors acknowledge that this is a "preliminary attempt" to develop the forward model and the results to this look point look interesting. After addressing the clarity issues below, this work could contribute helpfully to the scientific literature in this field.

Main concerns: 1. Not enough clarity and information in the description of the method - there are numerous sections (which I list individually below; points 4-8) where the paper is hard to follow. In particular, I am still unclear how the particle size distribution was retrieved from the radar observations. The only reference I can find in the paper says "We ... used the microphysical relations established in Section 3 to derive the PSD", but how exactly remains unspecified. Overall there is insufficient description to understand the method and/or to reproduce it.

2. The work nicely evaluates the difference between the forward-modelled PSDs for different particle types. However, for the retrieval of a PSD from the observed radar data some prior knowledge of the particle type/shape would be required. In this paper, the authors calculate six different size distributions - one for each particle type. Although the size distributions agree relatively well with aircraft measurements, there is an approximately order-of-magnitude difference in the absolute number concentration at different sizes. The authors do not explain how to overcome this need for a-priori information about particle type. Overcoming this issue would be required to perform the long-term statistical analysis they propose in the conclusions. Therefore I see difficulty
in finding real-world applicability of this work.

3. This is not the first attempt to determine ice cloud properties from Doppler spectral density data; however, the authors fail to detail the relative strengths and weaknesses of their method in relation to others already published. A discussion of how this work is similar or different to existing work (i.e. only one radar, but particle type is not known) is required in the introduction and/or later sections. Relevant references (and references therein):

for liquid: Kollias, P., Rémillard, J., Luke, E., and Szyrmer, W.: Cloud radar Doppler spectra in drizzling stratiform clouds: 1. Forward modeling and remote sensing applications, J. Geophys. Res.-Atmos., 116, D13201, https://doi.org/10.1029/2010JD015237, 2011

for ice: Barrett, A., Westbrook, C., Nicol, J., and Stein, T.: Rapid ice aggregation process revealed through triple-wavelength Doppler spectrum radar analysis, Atmos. Chem. Phys., 19, 5753–5769, 2019

Kneifel, S., Kollias, P., Battaglia, A., Leinonen, J., Maahn, M., Kalesse, H., & Tridon, F. (2016). First observations of triple-frequency radar Doppler spectra in snowfall: Interpretation and applications. Geophysical Research Letters, 43(5), 2225-2233.

Specific concerns: 4. page 9, line 224. Please state specifically which values of kurtosis and power-law prefactor used, or describe clearly which part of Hogan and Westbrook (2014) has been used (e.g. page/equation numbers)

5. page 11, line 256-268. From this paragraph is it unclear what is meant. One part says "the backscattering cross-sections are almost equal" (for the different particle types?), but later "it is crucial to choose shape parameters for ice particle types". I recommend rewriting this paragraph to clarify the intended meaning.

6. The largest problem I had with understanding related to the use of the PSDs. This is mostly covered in section 2.3.2 for the forward model. (The information for the inverse

AMTD
retrieval is completely missing) However, the details are also mixed with a lot of other content which made finding the relevant details difficult.

line 280 - please additionally define  $S_Z$  and  $V_f$  here.

line 284 - What do you mean by the "average PSD was used"

line 286 - how have you used a range of D\_e values (307-989)?

line 288 - initially the 35 dB range of reflectivity for the same particle size distribution was very confusing

line 291 - the reflectivity bias of 9.25 dB - is the difference explained by different masses or densities of the particles?

line 291 - what is assumed about the particle orientation during these calculations?

Figure 4 - the use of "...of single crystals" in the figure labels is very confusing (I was thinking: what is the size distribution of a single crystal?) and should be removed. I think you mean "of the different particle types" - but this would be clear from the figure legend and therefore does not need to be included on the figure axes.

line 296 - how have you determined the "Doppler spectral density width" - is this different from the "Doppler spectral density" mentioned previously?

line 310 - "it has the biggest" - what do you mean by "it"?

line 315-319 - please rewrite the entire sentence.

line 319 - what is "Doppler spectrum intensity magnitude"? please define

line 321 - I suggest adding a paragraph here describing what you are attempting in the upcoming paragraph. As of now, I do not understand how (or why) you are attempting to use the Doppler spectra of stellar plates, combined with the doppler spectra of aggregates of columns to derive the PSD of four other particle types - when earlier it was stated that the PSD from Figure 4a was used for all particle types.
line 327 - Further confusion comes when you state that "the retrieved PSDs ... are obviously larger than the original one". What do you mean by the "original one" here?

line 321-334 - The whole paragraph is complicated to follow and difficult to relate to figure 5. I suggest rewriting.

7. Section 3

line 346 - I suggest breaking the paragraph here, so that mode detail can be added about how the radar data and the inverted Doppler spectrum was used to determine the particle size distribution from the MMCR data. The text from line 346-352 is insufficient to explain what is the main benefit of your work. Additionally, the reference to section 3 (line 350) is incorrect, because the current section is section 3.

line 351 - what is the meaning of the "particle physical scale" in this sentence?

figure 6, caption - how did you determine the vertical air speed. This information needs to be added to the text. line 360 & 362 - what is the "PSD spectral width"? How is it defined and calculated?

8. A small table containing often-used symbols (De, Sz, Ze, sigma, etc.) would be useful

Minor comments:

9. line 33 - reference Liou (1986) incorrectly formatted

10. line 36 - please add a supporting reference for "most precipitation in China is related to theice phase process"

11. line 52 - "particle sizes are easier to calculate for liquid particles..." (+than for ice particles)

12. throughout paper - inconsistent use of "SZ" and "S\_z" (with subscript). Similarly for Dm, De and D\_m, D\_e (with subscript)

AMTD
- 13. line 140 MAGANO and Chung (1996) correct capitalisation
- 14. line 172 "calculation methods of calculation"
- 15. line 212 Hogan et al. reference missing year and missing from reference list
- 16. line 226 "different ice particles types of ice particles"
- 17. line 255 single sentence paragraph
- 18. line 285 "Doppler (+spectral) density"
- 19. figure 4b I suggest a different line style for aggregates as they use a different set of axes
- 20. line 324 "coordinate axes" -> "ordinate axes"
- 21. line 340 define "MMCR"
- 22. line 340/figure 6a velocity should be unfolded for the example plot
- 23. line 347 "inversed" -> "inverted"

24. figure 6 - suggest using same height axes (0-10 km) for both sub-panels. Also label the y-axis "Height" in both.

25. figure 7 - units of mm6 s m-4 are incorrect here

26. lines 411-415 - the averaging range appears to be in the melting level. This must affect the averaged quantities for the final analysis. A comment about this, or a choice of different height range is suggested.

27. line 426 & line 492 - I strongly disagree that rapid microphysical changes are required to produce precipitation. Precipitation could occur from a quasi steady state atmosphere. Without seeing the full time-height plot from the radar, I cannot determine whether the atmosphere really did vary rapidly during this time window - but I don't believe that this statement is justified.
28. figure 10 - suggest using same height axes (0-5 km) for both sub-panels. Also label the y-axis "Height" in both.

29. figure 10 caption - define "HVPS"

30. line 443 - is it possible to estimate how much LWC might have been missed because of the small particle problem?

31. line 446 - aliasing of what, where and how? Why is it relevant? Do you mean figure 6a, rather than 7a?

32. table 3 - what are the input and output units for IWC, Z e the equations given?

33. line 479-485 - conclusion point 2 is confusing

---

## Referee Comment (RC2) · Anonymous Referee #2 · 4 Dec 2019

The paper presents a methodology for the simulation of Doppler ice spectra for different ice habits. based on an hydrodynamic and scattering model (the self-similar Rayleigh Gans approximation). The topic is certainly timely and relevant. However I do not see how this paper is providing any novelty and any original contribution to the state of the art. First, radar spectra have been simulated for years both in rain and ice but the authors miss to cite any relevant reference in the field. Second, the simulations as provided clearly lack a lot of realism (no turbulence, no noise floor, ). The only simulated spectra that I can see are those produced in Fig.4, right panel. Such spectra have nothing to do with real spectra, do they? Why the authors are not plotting some of their spectra of Fig.6 for comparison? Therefore it does not make any sense to me

to attempt a retrieval (the pre-condition to that is that the forward model can reproduce the measured spectra). The results plotted in Fig.6 and 7 therefore make no sense to me. In addition to that it is totally unclear how the authors have accounted for the vertical velocity (the retrieval of the vertical velocity is not a trivial task!), for turbulence and spectral broadening, for attenuation (how can you retrieve IWC if you have not corrected for attenuation?) and for radar calibration. The comparison with the aircraft data is also extremely nebolous. Plot 11 with ice water contents all over the place clearly shows that there is something not right. At a certain point the authors are even extending their methodology to rain (again all previous work on the field is never mentioned), distracting the reader from the main topic. I do recommend the authors to properly refine/revisit their methodology and resubmit later.

---

## Author Comment (AC1) · 31 Dec 2019

**Answers to referee#1 on "Establishment and preliminary application**

**of forward modeling method for Doppler spectral density of ice**

**particles"**

Han Ding1, Liping Liu1

1 State Key Laboratory of Severe Weather, Chinese Academy of Meteorological Sciences, Beijing 100086, China

Correspondence: Liping Liu (liulp@cma.gov.cn)

The authors would like to thank the anonymous referee for the comments on the manuscript. The comments are so constructive and will help to sharpen and clarify the paper, all of them will be addressed in some manner. In the following, the comments are given in blue. The answers are given in normal black. The modified text in the manuscript is given in quotation marks.

**General response:**

After carefully considering all the comments of the reviewers, we have made the following modifications to the manuscript:

- We added some references to the studies of ice cloud properties using Doppler spectrum density data and discussed the differences between our work and theirs in the introduction and discussion part.
- With respect to the PSD retrieval method, we rewrote the relevant description in a clearer way.
- We realized that our simulation work wasn't good enough. We redid this part and the influence of turbulence and radar sensitivity on Doppler spectrum are evaluated.
- We are sorry for the inadequate description of the data post-processing, we have added the processing method to the data section.

**Response to main concerns:**

 Not enough clarity and information in the description of the method- there are numerous sections (which I list individually below; points 4-8) where the paper is hard to follow. In particular, I am still unclear how the particle size distribution was retrieved from the radar observations. The only reference I can find in the paper says "We ... used the microphysical relations established in Section 3 to derive the PSD", but how exactly remains unspecified. Overall there is insufficient description to understand the method and/or to reproduce it.

Actually, the Doppler spectrum observed by radar is defined as the function of the backscattering cross-section of the particles in the detection volume with respect to their fall velocity. The rationale of ice PSD retrieval is similar to the liquid drop size distribution. In fact, we consider the velocity corresponding to the leftmost point of the Doppler spectrum as the air speed at that layer, then the Doppler spectrum is shifted to correspond

to zero vertical air motion, the actual droplet size distribution (DSD) in stratiform precipitation can be retrieved form the Doppler spectrum under the assumption that turbulence effects on Doppler spectral density data are negligible. Similarly, for a certain type of ice crystal, the scattering cross-section and falling velocity is one-to-one with the particle scale, which means that we can match the radial velocity detected by radar to the particle size with the air motion removed (so Sz becomes a function of particle fall velocity, i.e. particle size). Since we can calculate the backscattering cross-section of particles of any size, then the PSD can be derived according to Eq. (1).

2. The work nicely evaluates the difference between the forward-modelled PSDs for different particle types. However, for the retrieval of a PSD from the observed radar data some prior knowledge of the particle type/shape would be required. In this paper, the authors calculate six different size distributions - one for each particle type. Although the size distributions agree relatively well with aircraft measurements, there is an approximately order-of-magnitude difference in the absolute number concentration at different sizes. The authors do not explain how to overcome this need for a-priori information about particle type. Overcoming this issue would be required to perform the long-term statistical analysis they propose in the conclusions. Therefore I see difficulty in finding real-world applicability of this work.

We totally agree that long-term observation and statistical analysis are needed to provide the priori information to bridge the gap between the retrieval PSD and Doppler spectrum, such as the specific distribution of different ice crystals of the same size (depends on the temperature and humidity...). As long as the proportions of different kinds of ice crystals are known, it is possible to use our method to retrieve the PSD accurately. To the author's best knowledge, such work has been done, see details in:

Baum, B. A., et al. (2005). "Bulk scattering properties for the remote sensing of ice clouds. Part I: Microphysical data and models." Journal of Applied Meteorology **44**(12): 1885-1895.

Because our field measurement only obtained one dataset of aircraft and radar joint observations, so our work is under the assumption that there is only one particle type exist in cloud, further verification requires the support of more data in the future. When there are more than one kind of particle exists, it is possible to convert their concentration ratio to the ratio of the Doppler spectrum generated by different kinds of particles based on the  $\sigma$ -D and vt-D relationship we established, and calculate the PSDs of various kind of particles. We tried to perform PSD inversion by using the ice crystal distribution model given by Baum et al., the results are shown in the figure below:

3. This is not the first attempt to determine ice cloud properties from Doppler spectral density data; however, the authors fail to detail the relative strengths and weaknesses of their method in relation to others already published. A discussion of how this work is similar or different to existing work (i.e. only one radar, but particle type is not known) is required in the introduction and/or later sections.

Thank you for the literature you recommended. We read each one carefully, and added the references to them in the introduction part. Additionally, an explanation of the strengths of our work and the differences between our work and that of others are added in the introduction and discussion section.

Line 54: "Many studies have focused on raindrop size distribution retrieval using Doppler spectral density data (Liu et al., 2014;Kollias et al., 2011;Gossard et al., 1997)"

Line70: "...Ze-IWC relationships. In order to obtain more detailed information, many studies have developed using dual-frequency and triple-frequency radar due to the selfrichness of these radar data. However, only the inversion of rain DSD or part of the ice PSD and the identification of some microphysical processes in the cloud can be achieved (Liu et al., 2019;Barrett et al., 2019;Kneifel et al., 2016;Kollias et al., 2011). So far, research on ice particle retrieval using MMCR Doppler spectral density in China has not been found. Therefore, we establish the relationship between ice particle microphysical parameters and Doppler spectral density data apply it to analyze microphysical and dynamic properties to verify the feasibility. At the same time, the results were compared with aircraft data in order to evaluate the performance of China's first cloud radar with a solid-state transmitter."

**Response to specific concerns:**

 page 9, line 224. Please state specifically which values of kurtosis and power-law prefactor used, or describe clearly which part of Hogan and Westbrook (2014) has been used (e.g. page/equation numbers)

We added the specific position of the parameters that we used in Hogan's work. (table 1)

5. page 11, line 256-268. From this paragraph is it unclear what is meant. One part says "the backscattering cross-sections are almost equal" (for the different particle types?), but later "it is crucial to choose shape parameters for ice particle types". I recommend rewriting this paragraph to clarify the intended meaning.

This part has been rewritten.

"The backscattering cross-sections of single ice particles and aggregates were calculated using the method mentioned above, and the results are shown in Fig. 3b. The values of backscattering cross-section of different types are close when the size of ice particles is small. For the same De, hexagonal columns have the largest backscattering cross-section, followed by stellar plates and sector plates, while the backscattering cross-section of hexagonal plates, two kinds of aggregates and ice spheres are relatively small and almost equal to each other. Additionally, we found that the the backscattering cross-sectional area has little correlation to the projected area of particles with the same volume. The backscattering cross-section only depends on the integral of the projected area in the electromagnetic wave propagation direction. For ice crystals of the same volume, if they have the same density, their projected area is large or small (their thickness is thick or thin), the differences of their backscattering cross-section are too small to neglect. Therefore, it is crucial to choose the mass parameters for ice particle types, which will significantly affect the calculation results. However, ice crystal shapes are very complex..."

6. The largest problem I had with understanding related to the use of the PSDs. This is mostly covered in section 2.3.2 for the forward model. (The information for the inverse retrieval is completely missing) However, the details are also mixed with a lot of other content which made finding the relevant details difficult.

line 280 - please additionally define Sz and Vf here.

line 284 - What do you mean by the "average PSD was used"

line 286 - how have you used a range of De values (307-989)?

line 288 - initially the 35 dB range of reflectivity for the same particle size distribution was very confusing

line 291 - the reflectivity bias of 9.25 dB - is the difference explained by different masses or densities of the particles?

line 315-319 - please rewrite the entire sentence.

Our initial purpose in choosing this size range of De values is to cover all types of ice crystals within the limits of the real physical size. After careful consideration, we adjusted the simulation work of and rewrite this part. We use the maximum diameter to describe the given PSDs and the effects of turbulence and radar sensitivity were evaluated after calculating the  $S_Z$ . The rewritten section 2.3.2 is in the following:

"which has been widely used then:

$$N(D) = N_0 exp(-\lambda D), \tag{6}$$

$$Z_e = \int_0^1 S_z(v_f) dv \tag{7}$$

Here, N(D) (m-3 mm-1) is the number of particles per unit volume per unit,  $N_0$  (m-3 mm-1) is the intercept parameter, and  $\lambda$  (mm-1) is the shape parameter. We use the Marshall-Palmer constants given by Platt (1997) at -10°C~-5°C, with N0 and  $\lambda$  values of 9560 m-3 mm-1 and 1.32 mm-1, respectively. For single ice crystals, we assume that D ranges from 100 to 5000 µm, and the average PSD was used to calculate the Doppler spectral density produced by four kinds of single ice crystals (based on Eq. (1)). The PSD used for Doppler spectrum simulation are shown in Fig. 4a. By using Eq. (7), the equivalent reflectivity values

generated by the four crystal types are 25.8, 24, 24.7 and 12.9 dBZ (hexagonal plates, hexagonal columns, sector-like plates, stellar crystals). Additionally, we calculated the Doppler spectrum affected by air turbulence at different intensities. According to Gossard et al. (1997), the convolution of Doppler spectrum in clear-air and the air turbulence can be written as:

$$S_{zT}(v_r) = \frac{1}{\sqrt{\pi}w_\sigma} \int_0^\infty S_z \left[ v_f(D) \right] exp \left[ \frac{-\left( v_r - v_f \right)^2}{w_\sigma^2} \right] dv_f , \qquad (8)$$

Here,  $S_{zT}$  and  $S_z$  represent the Doppler spectral density affected by turbulence and in clear-air, respectively.  $w_{\sigma}$  is the intensity of turbulence. As can be seen in Fig. 4b, air turbulence will broaden the Doppler spectrum while weakening its peak. The stronger the turbulence, the more severe the spectral distortion. For the turbulence of the same intensity, a narrower Doppler spectrum has more severe distortion. Because of the higher sensitivity of mode 3 of the radar, the sensitivity has a limited effect on the detection of Doppler spectrum.

Compare the Doppler spectra generated by different types of ice crystals with the same PSD in Fig. 4b, the width of the spectra is mainly inversely proportional to the rate at which the falling velocity increases with the particle scale. The faster the velocity increase with size, the wider the generated Doppler spectrum and vice versa. The value of  $S_Z$  is jointly determined by particle backscattering cross-section and  $\partial D / \partial V_f$  as shown in Eq. (1), which is proportional to the  $\sigma$  and inversely proportional to the rate of velocity change with particle size."

line 310 - "it has the biggest" - what do you mean by "it"?

line 321 - I suggest adding a paragraph here describing what you are attempting in the upcoming paragraph. As of now, I do not understand how (or why) you are attempting to use the Doppler spectra of stellar plates, combined with the doppler spectra of aggregates of columns to derive the PSD of four other particle types - when earlier it was stated that

the PSD from Figure 4a was used for all particle types.

line 327 - Further confusion comes when you state that "the retrieved PSDs ... are obviously larger than the original one". What do you mean by the "original one" here?

line 321-334 - The whole paragraph is complicated to follow and difficult to relate to figure 5. I suggest rewriting.

We have adjusted the work of this part and rewrote the whole paragraph, please check it below.

---

## Author Comment (AC2) · 31 Dec 2019

**Answers to referee#2 on "Establishment and preliminary application of forward modeling method for Doppler spectral density of ice particles"**

Han Ding[1], Liping Liu[1]

1 State Key Laboratory of Severe Weather, Chinese Academy of Meteorological Sciences, Beijing 100086, China

**Correspondence**: Liping Liu (liulp@cma.gov.cn)

The authors would like to thank the anonymous referee for the comments on the manuscript. The comments are constructive and will help to sharpen and clarify the paper, all of them will be addressed in some manner. In the following, the comments are given in blue. The answers are given in normal black. The modified text in the manuscript is given in quotation marks.

**General response:**

After carefully considering all the comments of the reviewers, we have made the following modifications to the manuscript:

- We added some references to the studies of ice cloud properties using Doppler spectrum density data and discussed the differences between our work and theirs in the introduction and discussion part.
- With respect to the PSD retrieval method, we rewrote the relevant description in a clearer way.
- We realized that our simulation work wasn't good enough. We redid this part and the influence of turbulence and radar sensitivity on Doppler spectrum are evaluated.
- We are sorry for the inadequate description of the data post-processing, we have added the processing method to the data section.

The paper presents a methodology for the simulation of Doppler ice spectra for different ice habits. based on an hydrodynamic and scattering model (the self-similar Rayleigh Gans approximation). The topic is certainly timely and relevant. However I do not see how this paper is providing any novelty and any original contribution to the state of the art. First, radar spectra have been simulated for years both in rain and ice but the authors miss to cite any relevant reference in the field. Second, the simulations as provided clearly lack a lot of realism (no turbulence, no noise floor, ). The only simulated spectra that I can see are those produced in Fig.4, right panel. Such spectra have nothing to do with real spectra, do they? Why the authors are not plotting some of their spectra of Fig.6 for comparison? Therefore it does not make any sense to me to attempt a retrieval (the pre-condition to that is that the forward model can reproduce the measured spectra). The results plotted in Fig.6 and 7 therefore make no sense to me. In addition to that it is totally unclear how the authors have accounted for the vertical velocity (the retrieval of the vertical velocity is not a trivial task!), for turbulence and spectral broadening, for attenuation (how can you retrieve IWC if you have not corrected for attenuation?) and for radar calibration. The comparison with the aircraft data is also extremely nebolous. Plot 11 with ice water contents all over the place clearly

shows that there is something not right. At a certain point the authors are even extending their methodology to rain (again all previous work on the field is never mentioned), distracting the reader from the main topic. I do recommend the authors to properly refine/revisit their methodology and resubmit later.

**Response to specific concerns:**

**Novelty and original contribution:**

We have established a method that can retrieve the complete PSD of ice crystals for the first time, and evaluated the feasibility using the data observed by China's first radar with a solid-state transmitter. According to the results, as long as the proportions of different kinds of ice crystals are known, it is possible to use our method to retrieve the PSD accurately. Because our field measurement only obtained one dataset of aircraft and radar joint observations, to compare with the results of the aircraft, our work is under the assumption that there is only one particle type exist in cloud, further verification requires the support of more data in the future (long-term observation and statistical analysis are needed to provide the priori information). When there are more than one kind of particle exists, it is possible to convert their concentration ratio to the ratio of the Doppler spectrum generated by different kinds of particles based on the $\sigma$-D and $v_t$-D relationship we established, and calculate the PSDs of various kind of particles. We tried to perform PSD inversion by using the ice crystal distribution model given by Baum et al.(Baum et al., 2005), the results are shown in the figure below:

[Figure]

**Introduction**:

We have added references to the related research in this field. Additionally, an explanation of the strengths of our work and the differences between our work and that of others are added in the introduction and discussion section.

Line 54: "Many studies have focused on raindrop size distribution retrieval using Doppler spectral density data (Liu et al., 2014;Kollias et al., 2011;Gossard et al., 1997)"
Line70: "…$Z_e$-IWC relationships. In order to obtain more detailed information, many studies have developed using dual-frequency and triple-frequency radar due to the self-richness of these radar data. However, only the inversion of rain DSD or part of the ice PSD and the identification of some microphysical processes in the cloud can be achieved (Liu et al.,

2019;Barrett et al., 2019;Kneifel et al., 2016;Kollias et al., 2011). So far, research on ice particle retrieval using MMCR Doppler spectral density in China has not been found. Therefore, we establish the relationship between ice particle microphysical parameters and Doppler spectral density data apply it to analyze microphysical and dynamic properties to verify the feasibility. At the same time, the results were compared with aircraft data in order to evaluate the performance of China's first cloud radar with a solid-state transmitter."

**Determine the vertical velocity and the retrieval method:**

Actually, the Doppler spectrum observed by radar is defined as the function of the backscattering cross-section of the particles in the detection volume with respect to their fall velocity. The rationale of ice PSD retrieval is similar to the liquid drop size distribution. In fact, we consider the velocity corresponding to the leftmost point of the Doppler spectrum as the air speed at that layer, then the Doppler spectrum is shifted to correspond to zero vertical air motion, the actual droplet size distribution (DSD) in stratiform precipitation can be retrieved form the Doppler spectrum under the assumption that turbulence effects on Doppler spectral density data are negligible. Similarly, for a certain type of ice crystal, the scattering cross-section and falling velocity is one-to-one with the particle scale, which means that we can match the radial velocity detected by radar to the particle size with the air motion removed (so Sz becomes a function of particle fall velocity, i.e. particle size). Since we can calculate the backscattering cross-section of particles of any size, then the PSD can be derived according to Eq. (1).

**Doppler spectrum simulation:**

The main purpose of our simulation work using the same certain PSD to simulate the Doppler spectrum is to show that even if the same PSD, if the ice crystal habit is different, the Doppler spectrum they produce will also very different. After careful consideration, we adjusted the simulation work of and rewrite this part. We use the maximum diameter to describe the given PSDs and the effects of turbulence and radar sensitivity were evaluated after calculating the $S_Z$. The rewritten section 2.3.2 is in the following:
"which has been widely used then:

$$N(D) = N_0 exp(-\lambda D), \tag{6}$$

$$Z_e = \int_0^v S_z(v_f)dv \tag{7}$$

Here, $N(D)$ $(m^{-3} mm^{-1})$ is the number of particles per unit volume per unit, $N_0$ $(m^{-3} mm^{-1})$ is the intercept parameter, and $\lambda$ $(mm^{-1})$ is the shape parameter. We use the Marshall-Palmer constants given by Platt (1997) at $-10°C \sim -5°C$, with $N_0$ and $\lambda$ values of 9560 $m^{-3} mm^{-1}$ and 1.32 $mm^{-1}$, respectively. For single ice crystals, we assume that D ranges from 100 to 5000 μm, and the average PSD was used to calculate the Doppler spectral density produced by four kinds of single ice crystals (based on Eq. (1)). The PSD used for Doppler spectrum simulation are shown in Fig. 4a. By using Eq. (7), the equivalent reflectivity values generated by the four crystal types are 25.8, 24, 24.7 and 12.9 dBZ (hexagonal plates, hexagonal columns, sector-like plates, stellar crystals). Additionally, we calculated the

Doppler spectrum affected by air turbulence at different intensities. According to Gossard et al. (1997), the convolution of Doppler spectrum in clear-air and the air turbulence can be written as:

$$S_{zT}(v_r) = \frac{1}{\sqrt{\pi} w_\sigma} \int_0^\infty S_z[v_f(D)] exp\left[\frac{-(v_r - v_f)^2}{w_\sigma^2}\right] dv_f ,\qquad (8)$$

Here, $S_{zT}$ and $S_z$ represent the Doppler spectral density affected by turbulence and in clear-air, respectively. $w_\sigma$ is the intensity of turbulence. As can be seen in Fig. 4b, air turbulence will broaden the Doppler spectrum while weakening its peak. The stronger the turbulence, the more severe the spectral distortion. For the turbulence of the same intensity, a narrower Doppler spectrum has more severe distortion. Because of the higher sensitivity of mode 3 of the radar, the sensitivity has a limited effect on the detection of Doppler spectrum.

[Figure]

Figure 4. (a) Exponential distribution used in this paper. (b) The Doppler spectral density calculated from the exponential distribution without turbulence (the solid line) and at different turbulence intensities (the dashed line). Radar sensitivity at 5 km of M3 are marked by a solid black line.

Compare the Doppler spectra generated by different types of ice crystals with the same PSD in Fig. 4b, the width of the spectra is mainly inversely proportional to the rate at which the falling velocity increases with the particle scale. The faster the velocity increase with size, the wider the generated Doppler spectrum and vice versa. The value of $S_Z$ is jointly determined by particle backscattering cross-section and $\partial D/\partial V_f$ as shown in Eq. (1), which is proportional to the $\sigma$ and inversely proportional to the rate of velocity change with particle size."

[Figure]

Figure 5. PSDs retrieved from Doppler spectra affected by turbulence shown in fig. 4b. The solid black line is the PSD given by Platt (1977) at −10°C~−5°C (same as fig. 4a).

 "To further study the effect of turbulence on inversion of the PSD, the affected Doppler spectra were used to invert the new PSD and compared with the original given PSD. According to fig.5, the new PSDs are significantly wider than the original, an overestimate of the number of particles was occurred when D is small, and an underestimate of the particle number was occurred at the large D. It can be easily seen that stronger turbulence has a greater impact on the inverted PSD, which will cause the particle number deviate from the true value seriously. Moreover, different types of ice crystals have varying degrees of sensitivity to turbulence. Compared with sector crystals and stellar crystals, hexagonal plates and hexagonal columns are less affected by turbulence."

**Data processing:**

Actually, all the spectrum data is quality controlled before use. We have calculated the noise level and removed the non-meteorological echo, and calculated the attenuation coefficient using the inverted rain droplet size distribution and corrected it. Since the stable stratiform precipitation occurred during the observation and the width of Doppler spectrum was wide, the influence of turbulence was ignored during the inversion.

We will add the details about data post-processing in the manuscript:

"The raw Doppler spectral density data from the four work modes are post-processed and used to recalculated reflectivity, retrieve the vertical air motion. The data post-processing includes quality control (QC), merging for Doppler spectra and recalculating Doppler moments (Liu, et al, 2019). QC for Doppler spectra includes dealiasing singly wrapped aliased Doppler spectral density data and detecting and removing artifacts produced by pulse compression. After QC and merging, we directly estimated the vertical air velocity using the velocity bin of small particles such as liquid droplets and small ice crystals assuming that these particles can be considered tracers of clear-air motion in the measured spectra, we conducted attenuation

correction of Doppler spectral density bin by bin from the first range to the end. We calculated the attenuation coefficient from liquid drop size distribution (DSD)."

References:

Barrett, A. I., Westbrook, C. D., Nicol, J. C., and Stein, T. H.: Rapid ice aggregation process revealed through triple-wavelength Doppler spectrum radar analysis, Atmospheric Chemistry and Physics, 19, 5753-5769, 2019.

Baum, B. A., Heymsfield, A. J., Yang, P., and Bedka, S. T.: Bulk scattering properties for the remote sensing of ice clouds. Part I: Microphysical data and models, Journal of Applied Meteorology, 44, 1885-1895, 2005.

Gossard, E. E., Snider, J. B., Clothiaux, E. E., Martner, B., Gibson, J. S., Kropfli, R. A., and Frisch, A. S.: The Potential of 8-mm Radars for Remotely Sensing Cloud Drop Size Distributions, Journal of Atmospheric and Oceanic Technology, 14, 76-87, 10.1175/1520-0426(1997)014<0076:tpomrf>2.0.co;2, 1997.

Kneifel, S., Kollias, P., Battaglia, A., Leinonen, J., Maahn, M., Kalesse, H., and Tridon, F.: First observations of triple-frequency radar Doppler spectra in snowfall: Interpretation and applications, Geophysical Research Letters, 43, 2225-2233, 2016.

Kollias, P., Rémillard, J., Luke, E., and Szyrmer, W.: Cloud radar Doppler spectra in drizzling stratiform clouds: 1. Forward modeling and remote sensing applications, Journal of Geophysical Research: Atmospheres, 116, 2011.

Liu, L., Xie, L., and Cui, Z.: Examination and application of Doppler spectral density data in drop size distribution retrieval in weak precipitation by cloud radar, Chinese Journal of Atmospheric Sciences (in Chinese), 38, 223-236, 10.3878/j.issn.1006-9895.2013.12207, 2014.

Liu, L., Ding, H., Dong, X., Cao, J., and Su, T.: Applications of QC and Merged Doppler Spectral Density Data from Ka-Band Cloud Radar to Microphysics Retrieval and Comparison with Airplane in Situ Observation, Remote Sensing, 11, 1595, 2019.

Platt, C. M. R.: A parameterization of the visible extinction coefficient of ice clouds in terms of the ice/water content, Journal of the atmospheric sciences, 54, 2083-2098, 1997.